# Role of Oxidative Stress in Tuberculosis Meningitis Infection in Diabetics

**DOI:** 10.3390/biomedicines11092568

**Published:** 2023-09-19

**Authors:** Inesa Navasardyan, Stephanie Yeganyan, Helena Nguyen, Payal Vaghashia, Selvakumar Subbian, Vishwanath Venketaraman

**Affiliations:** 1College of Osteopathic Medicine of the Pacific, Western University of Health Sciences, Pomona, CA 91766, USA; inesa.navasardyan@westernu.edu (I.N.); stephanie.yeganyan@westernu.edu (S.Y.); helena.nguyen@westernu.edu (H.N.); payal.vaghashia@westernu.edu (P.V.); 2Public Health Research Center, New Jersey Medical School, Rutgers University, Newark, NJ 07103, USA; subbiase@njms.rutgers.edu

**Keywords:** *Mycobacterium tuberculosis*, tuberculosis meningitis, diabetes, oxidative stress

## Abstract

Tuberculosis meningitis (TBM) is a result of the invasion of the meninges with the bacilli of *Mycobacterium tuberculosis* (Mtb), leading to inflammation of the meninges around the brain or spinal cord. Oxidative stress occurs when the body’s cells become overwhelmed with free radicals, particularly reactive oxygen species (ROS). ROS plays a significant role in the pathogenesis of TBM due to their toxic nature, resulting in impairment of the body’s ability to fight off infection. ROS damages the endothelial cells and impairs the defense mechanisms of the blood–brain barrier (BBB), which contributes to CNS susceptibility to the bacteria causing TBM. Diabetes mellitus (DM) is a common condition that is characterized by the impairment of the hormone insulin, which is responsible for modulating blood glucose levels. The increased availability of glucose in individuals with diabetes results in increased cellular activity and metabolism, leading to heightened ROS production and, in turn, increased susceptibility to TBM. In this review, we summarize our current understanding of oxidative stress and its role in both TBM and DM. We further discuss how increased oxidative stress in DM can contribute to the likelihood of developing TBM and potential therapeutic approaches that may be of therapeutic value.

## 1. Introduction

Tuberculous meningitis (TBM) is an extrapulmonary manifestation of *Mycobacterium tuberculosis* (Mtb) infection of the central nervous system (CNS), resulting in non-suppurative inflammation of the meninges of the brain or spinal cord [1,2]. In humans, TBM is thought to be secondary to pulmonary tuberculosis (TB), the most predominant form of TB that begins with inhalation of Mtb into the lungs, where it infects alveolar macrophages, neutrophils, and dendritic cells, and continues to replicate [3]. Impaired anti-mycobacterial immune responses can lead to a severe form of cavitary TB that can further result in the dissemination of Mtb to the CNS [4]. Experimental models have proposed several mechanisms that promote entry of Mtb across the blood–brain barrier (BBB), allowing access to the CNS. Research shows that invasion of the BBB by Mtb results in cytoskeletal actin rearrangements of brain cells [5]. Furthermore, specific genes were found to be upregulated during Mtb invasion of the BBB, suggesting the role of possible Mtb virulence factors in transmission [5]. In addition, Mtb-infected macrophages can penetrate the BBB, with simultaneous elevation in the levels of vascular endothelial growth factor (VEGF), suggesting VEGF’s possible role in Mtb invasion [6]. Although it is difficult to quantify the true global burden of TBM due to poor diagnostics and surveillance, research suggests that TBM contributes to about 5–10% of all extrapulmonary tuberculosis (TB) cases, and approximately 1–10% of TB patients develop TBM, strongly depending on the local prevalence [7]. Among the major factors that affect TBM diagnosis and prognosis are age and HIV prevalence. Over 80% of individuals with TBM were HIV-infected in a high HIV prevalence setting [8]. In children, a higher rate of TBM can be due to a weakened or immature immune system. Researchers found that 30% of children with TBM were in contact with an individual with pulmonary Mtb, in contrast to only 13% of adults with TBM [9]. Nevertheless, TBM remains the most fatal manifestation among all forms of tuberculosis, with a significant frequency of mortality and morbidity, reaching approximately 40% [10,11].

Diabetes mellitus (DM) is a chronic condition in which the body fails to produce and respond to insulin, resulting in elevated glucose levels [12]. Common characteristics of individuals living with DM include thirst, frequent urination, and weight loss. Type 1 diabetes (T1D) is characterized by cell-mediated auto-immune destruction of the pancreatic β-cells responsible for producing insulin. Conversely, type 2 diabetes (T2D) is attributed to insulin resistance and reduced insulin secretion [13]. Currently, there are 29.1 million individuals living with DM in the United States, with T2D being more prevalent than T1D [14]. Both T1D and T2D are associated with genetic predispositions and lifestyle factors [15]. Current treatment strategies for DM include lifestyle changes and insulin therapy, which works to mimic the body’s innate insulin secretion. Depending on the stage of the disease, exogenous insulin can be used for both T1D and T2D and has been shown to reduce complications associated with DM, such as neuropathy, retinopathy, and kidney disease [16]. For patients suffering from T2D, lifestyle intervention and metformin, a medication that increases insulin sensitivity, were shown to decrease the incidence of diabetes by 58% and 31%, respectively [17]. More targeted approaches, such as exosome therapy, are being considered as future treatments for DM [18].

Oxidative stress occurs when there exists an imbalance between the production of reactive oxygen species (ROS) and the capacity of antioxidants to remove such toxic substances from the body [19]. ROS are natural byproducts of physiological processes in the body important for cell signaling; however, it is when the production of ROS exceeds the ability of the body to effectively remove such substances that the body experiences a state of oxidative stress, leading to cell tissue damage [19]. The increase in cellular activity and metabolism observed in individuals with diabetes renders them particularly susceptible to increased oxidative stress [20]. The role of oxidative stress in the pathogenesis of TBM has been studied. The aim of this review is to provide several mechanisms in which increased oxidative stress in diabetic patients may increase one’s likelihood of developing TBM.

## 2. TB and Diabetes

Diabetes can have a significant impact on Mtb infection, including increased susceptibility to active TB infection, altered immune responses, and poor treatment outcomes. Multiple studies have shown that asymptomatic infection with Mtb, termed latent TB infection (LTBI), is found in higher numbers among individuals with pre-diabetes and T2D compared to those without [21,22]. Diabetes is also a well-studied risk factor for the progression from LTBI to active TB disease [23,24]. T2D predisposes individuals to develop active TB and, thus, as the prevalence of T2D increases, the global burden of TB is also thought to increase. Recent evidence suggests that, in addition to hyperglycemia, hypercholesterolemia, and elevated triglyceride levels associated with DM, diabetes is also a main contributor to the increased susceptibility to TB [24].

Chronic inflammation and altered immune defense mechanisms in diabetics may compromise the ability of the host immune system to combat Mtb infection [25]. Furthermore, the ability of immune cells to recognize and destroy Mtb may be impaired, allowing the bacteria to replicate and manifest elsewhere, such as in the meninges in the case of TBM. For instance, there exists an inverse relationship between the activity of natural killer (NK) cells and hyperglycemia, wherein an increase in blood sugar levels is associated with an observed decrease in NK cell function in the innate and adaptive immune response against infection [26]. Moreover, another study demonstrated that endoplasmic reticulum (ER) stress as a result of uncontrolled hyperglycemia led to the downregulation of natural-killer receptor group 2, member D (NKG2D), a well-studied NK cell-activating receptor [27]. Induced hyperglycemia was found to impair neutrophil degranulation, a mechanism of pathogen clearance involving extracellular destruction [28]. Moreover, long-term high-glucose concentrations sensitize macrophages to cytokine stimulation while reducing the phagocytic and bactericidal function of macrophages essential in fighting pathogens [29]. Lastly, the role of cytokines in the increased susceptibility of diabetics to infection has been studied. Upregulation of pro-inflammatory IL-6 and TNF-α has been observed in diabetics, whereas there is a reduction in the circulating levels of anti-inflammatory IL-10 [30,31]. Thus, cytokine dysregulation in diabetic individuals, involving an increase in pro-inflammatory cytokines along with a decrease in anti-inflammatory cytokines, results in a chronic inflammatory cellular state, increasing one’s susceptibility to infection (Figure 1).

Multidrug-resistant TB (MDR-TB) is characterized by the resistance to at least isoniazid and rifampin, two potent drugs used in the treatment of TB. Diabetes, in particular T2D, is associated with a two-fold increased risk of MDR-TB [32]. Whole-genome sequencing of clinical isolates of Mtb revealed several mutations in individuals with T2D conferring resistance to isoniazid, ethionamide, fluoroquinolone, and rifampicin [33]. Interestingly, the serum concentrations of isoniazid and pyrazinamide were reduced in TB patients with T2D compared to those without T2D, which may be attributed to the generally higher BMI observed in T2D patients and/or the increased metabolic rate of anti-TB drugs in diabetics [34,35].

The relationship between DM and TB infection has been well studied. In this review, however, we provide novel insight into the potential mechanisms that predispose diabetic individuals to the extrapulmonary manifestation of TB infection into TBM.

## 3. Pathogenesis of TBM in Humans

Following Mtb infection of the cells of the CNS, slow-progressive meningitis develops with clinical symptoms such as headaches, vomiting, fever, and neck stiffness, indistinguishable from other meningitis types [36]. If untreated, TBM progresses to severe clinical conditions, including unconsciousness, focal neurological deficits, cranial nerve palsies, seizures, raised intracranial pressure, and hemiparesis. These symptoms are driven by the exacerbated inflammation caused by Mtb infection [36]. Elevated levels of pro-inflammatory cytokines, including tumor necrosis factor-alpha (TNF-α) released by infected immune cells into the CNS, contribute to the local and systemic inflammation in TBM, with the fifth and third cranial nerves being involved in approximately 50% of patients [37]. In advanced TBM, about 10% of patients report either hemiplegia or paraplegia, and death is almost inevitable at this stage, without proper therapeutic interventions [38]. Pleocytosis with lymphocyte predominance (150–1000 leukocytes/µL), enriched with neutrophils and lymphocytes, low glucose levels with the cerebrospinal fluid (CSF) to plasma glucose ratios of <0.5, and high protein content (0.8–2.0 g/dL) are some of the essential laboratory findings in the CSF of patients with TBM [10]. Furthermore, in patients with human immunodeficiency virus (HIV) co-infection, TBM is characterized by the absence of mononuclear leukocytes and the prominent presence of neutrophils (>1000 cells/µL) in the CSF, mimicking acute pyogenic bacterial meningitis [10]. Elevated levels of inflammatory cytokines are commonly noted in the plasma and CSF of patients with TBM. Radiographically, patients with TBM show basal meningeal exudates, infarction, tuberculomas, and hydrocephalus [10].

## 4. Oxidative Stress in the Pathogenesis of TBM

Increased production of reactive oxygen species (ROS) has been found to be intimately involved in the pathogenesis of TB and TBM [4,39]. Upon exposure to Mtb infection, the host immune system activates several defense mechanisms to control the infection, one of which includes the generation of ROS. Although ROS is initially produced to protect against infection, its overproduction leads to host tissue damage, including brain tissue, contributing to the pathogenesis of TBM. Increased ROS levels have been shown to impair the host immune response against Mtb infection and, in turn, allow infection to ensue in other parts of the body, including the brain [40]. Biomarkers identified from cerebrospinal fluid (CSF) samples from individuals with TBM demonstrated increased inflammation, disruption of the BBB, and impairment of amyloid-β (Aβ) metabolism [41].

Cellular damage and inflammation play a significant role in the pathogenesis of TBM. Both pro- and anti-inflammatory cytokines have been shown to be induced in individuals with TBM; however, it is the imbalance between the two that is responsible for the pathogenesis and prognosis of TBM [4]. Damage to cellular structures including lipids, proteins, and nucleic acids induced by excessive ROS production may lead to chronic inflammation and, in turn, increases the likelihood of TBM [42].

ROS may also have a significant impact on the integrity of the BBB. The lining of the BBB acts to protect the CNS by regulating the passage of molecules between the bloodstream and the brain. The BBB consists of tightly connected cells, including endothelial cells, astrocytes and pericytes. Increased ROS production and the resulting oxidative stress have been shown to damage endothelial cells crucial for the integrity of the BBB, resulting in increased permeability of the BBB to infection, including Mtb [43]. Subsequent impairment of tight junctions between endothelial cells also contributes to the dysregulation of the BBB. Moreover, ROS-induced production of pro-inflammatory stimuli, such as the redox-sensitive transcription factor NF-kB, can further impair the BBB by promoting leukocyte adhesion and extravasation, leading to increased neuroinflammation [43]. Furthermore, the BBB employs antioxidant defense mechanisms, such as superoxide dismutase (SOD) and glutathione peroxidase (GPx), to neutralize ROS and maintain redox homeostasis; however, in conditions of ROS overproduction, these antioxidants become overwhelmed and lead to subsequent damage of the BBB [44]. Lastly, increased ROS production may result in the upregulation of matrix metalloproteinases (MMPs), which are enzymes found to degrade components of the extracellular matrix, including those in the BBB, further disrupting the integrity of tight junctions and basal lamina of the BBB [43]. In summary, the accumulation of ROS leads to increased permeability of the BBB, allowing the infiltration of harmful and infectious agents such as Mtb, leading to the culmination of Mtb infection in the meninges. Thus, the increase in ROS production seen in diabetics may significantly increase an individual’s susceptibility to TBM.

Impaired metabolism of Aβ peptide metabolism and its resulting aggregation is a well-known pathological hallmark of Alzheimer’s disease. However, emerging evidence suggests that Aβ aggregation and deposition may also be implicated in the pathogenesis of diabetes and related neurodegenerative and neurovascular changes [45]. Hyperglycemia is associated with increased amyloid precursor protein (APP), leading to an increase in Aβ peptide production and increased permeability through endothelial tight junctions [46]. Furthermore, insulin resistance, a hallmark of T2DM, has been shown to impair Aβ clearance through the BBB, resulting in Aβ accumulation in the brain [46]. Oxidative stress, such as that seen in diabetic patients, further promotes Aβ aggregation [47]. Thus, one may argue that the increase in Aβ peptides observed in CSF samples of diabetic individuals suffering from TBM may be partly attributed to oxidative stress.

## 5. Oxidative Stress in Diabetes

Oxidative stress is implicated in the onset of insulin resistance due to the impairment of insulin signaling pathways, as well as in β-cell dysfunction and apoptosis, reducing the number of functional pancreatic β-cells [48]. Persistent hyperglycemia makes those with diabetes particularly susceptible to oxidative stress due to enhanced ROS production as well as impaired antioxidant defense mechanisms [49]. Thus, therapeutic interventions targeting oxidative stress in diabetes management through modulating ROS production or antioxidant supplementation have gained attention (Figure 2). 

### 5.1. Increased ROS Production in Diabetics

Diabetes may lead to an overproduction of ROS primarily through increased glucose auto-oxidation, activation of the polyol pathway, and the generation of advanced glycation end products (AGEs) [50]. Hyperglycemia is considered a significant source of ROS in the body since the increased availability of glucose contributes to an elevated rate of cellular processes involving glucose and, thus, an increase in ROS generation [50]. Glucose auto-oxidation is described as the spontaneous oxidation of glucose in the presence of oxygen or other electron acceptors. Under normal conditions, glucose exists in equilibrium with its enediol; however, under conditions of glucose auto-oxidation, an enediol radical is formed [51]. Glucose auto-oxidation is a significant source of free radical formation, such as hydrogen peroxide (H_2_O_2_) and superoxide anion [50]. The polyol pathway, also termed the sorbitol pathway, involves the conversion of glucose to sorbitol and then to fructose, using the enzyme aldose reductase (AR). This pathway is another mechanism through which hyperglycemia can lead to ROS generation [52]. Overactivity of AR under hyperglycemia conditions subjects’ diabetics to increased activity of the polyol pathway and, thus, increased ROS generation [52]. The conversion of glucose to sorbitol involves the consumption and depletion of NADPH, a necessary component of the host antioxidant defense system against ROS [52]. Lastly, the increased formation of AGEs in diabetes has been shown to increase ROS [53]. AGEs are modified proteins or lipids that have been non-enzymatically glycated and oxidized in response to glucose exposure. AGEs may interact with receptors for advanced glycation end products (RAGE) to increase ROS production via NADPH oxidases within the mitochondria, resulting in increased inflammation and tissue damage [54].

### 5.2. Impaired Antioxidant Defense Mechanisms in Diabetics

Antioxidants play a crucial role in the response against the production of ROS by effectively counteracting the detrimental effects of free radicals and mitigating the onset of oxidative stress. Consequently, impaired functionality of the antioxidant defense mechanism results in the accumulation of harmful species and, in turn, increased oxidative stress and damage. Due to the high metabolic activity in the brain, it relies on antioxidant defense mechanisms to exert neuroprotective effects by scavenging ROS and defending against oxidative damage, yet their efficiency is compromised in diabetics [55]. Diabetes is associated with reduced levels of antioxidants, such as glutathione, and antioxidant enzymes, such as glutathione reductase (GR), superoxide dismutase (SOD), and catalase (CAT) [56]. Another study revealed a positive correlation between GR, CAT, and SOD with malondialdehyde (MDA), a marker of oxidative stress [57].

Levels of glutathione have been reported to be decreased in individuals with diabetes compared to non-diabetic individuals [58,59]. The active and reduced form of glutathione (GSH) plays a crucial role in ROS scavenging, whereas the oxidized form of glutathione (GSSG) is often increased in conditions of oxidative stress. Thus, a reduced plasma GSH/GSSG ratio, such as that seen in diabetic individuals, may be used as an indicator of oxidative stress [60]. One study attributed GSH deficiency in patients with T2DM to the limited availability of glutathione precursors and, upon dietary administration of the GSH precursor amino acids cysteine and glycine, GSH levels were restored, and levels of oxidative stress and oxidative damage were reduced [58]. Another study using a diabetic mouse model demonstrated that liposomal glutathione (L-GSH) supplementation used in combination with rifampicin improved the treatment response against Mtb infection [61]. Moreover, GR is the enzyme responsible for maintaining glutathione in its reduced and active form by catalyzing the reduction of GSSG with the use of an NADPH electron donor. The decrease in GR activity in diabetics results in increased levels of GSSG and, thus, a diminishment in the levels of GSH and its ability to combat oxidative damage [62].

SODs are a group of enzymes that defend against ROS via the dismutation of superoxide radicals (O2-) to molecular oxygen and H_2_O_2_. Superoxide radicals produced in excess amounts lead to cell damage and, therefore, must be regulated by the action of SOD [63]. The expression and activity of the several isoforms of SOD varies in diabetes. Cytosolic CuZn-SOD (SOD1) and extracellular CuZn-SOD (SOD3) were downregulated in KK/Ta-Akita diabetic mouse models susceptible to diabetic nephropathy, whereas no significant change in mitochondrial Mn-SOD was observed [64]. Another study found reduced serum levels of extracellular SOD to be associated with polyneuropathy in individuals with recent-onset diabetes [65].

CAT is an antioxidant enzyme that catalyzes the reduction of H_2_O_2_ into water and molecular oxygen, protecting the body from oxidative damage. Although CAT activity in the brain is lower compared to other tissues and organs, such as the liver and kidney, it is still regarded as an important antioxidant necessary for proper brain function [66]. Inhibition of CAT activity is associated with enhanced cytotoxicity and increased levels of ROS, namely the accumulation of hydrogen peroxide that is responsible for compromised neurological function [66]. Low levels of catalase activity and the subsequent accumulation of H_2_O_2_ contribute to the pathogenesis of T2DM and related diabetic complications [67].

### 5.3. Role of Inflammation in Oxidative Stress and Diabetes

There also exists an intricate relationship between inflammation and oxidative stress in the context of diabetes, wherein diabetes-mediated inflammation triggers the production of pro-inflammatory molecules as well as the production of ROS by immune cells, aimed at combating pathogens [68]. Although these events are initially protective, chronic inflammation in diabetes leads to sustained ROS production, causing cellular damage and depletion of antioxidant defenses [69]. Moreover, the presence of excess adipose tissue in patients with T2D leads to the release of pro-inflammatory molecules, such as IL-6 and TNF-α, intensifying oxidative stress [70]. Additionally, the hexosamine pathway demonstrates the effects of hyperglycemia in the alteration of gene expression and contribution to diabetic complications, including tissue-type plasminogen activator inhibitor-1 (PAI-1) and transforming growth factor-β1 (TGF-β1) [68]. PAI-1 and TGF-β1 play a crucial role in the production of ROS and promotion of oxidative stress in diabetic nephropathy and neuropathy, respectively [68]. Finally, oxidative stress resulting from a pro-inflammatory state has been shown to impair insulin secretion and beta-cell function, ultimately worsening hyperglycemia [71]. Inflammation clearly plays a pivotal role in the advancement of oxidative stress, making it a potential therapeutic target in diabetic individuals, as observed using metformin.

## 6. Effects of Anti-Diabetic Agents on Oxidative Stress

DM is commonly managed through lifestyle modifications and medications. Metformin is the currently recommended first-line treatment of T2DM and exerts its effects by inhibiting hepatic gluconeogenesis and increasing insulin sensitivity, resulting in enhanced glucose uptake and reduced blood glucose levels [72,73]. One study demonstrated that rats treated with metformin exhibited an associated decrease in oxidative stress markers, such as conjugated dienes and thiobarbituric acid-reactive substances (TBARS), when compared to control groups [74].

Metformin has been proposed to lower oxidative stress through its inhibition of the mitochondrial electron transport chain (ETC) [72]. The ETC comprises four complexes that work together to generate an electrochemical gradient that ultimately results in the generation of ATP. However, the ETC can work in reverse, a process that has been proposed to be inhibited by metformin [72]. By inhibiting complex, I (NADH: ubiquinone oxidoreductase), metformin directly lowers mitochondrial-induced production of ROS, decreasing oxidative stress [72] (Figure 3). 

As previously mentioned, increased AGEs in diabetic patients can lead to increased ROS production. Metformin also plays a protective role against AGEs. In a three-month study of T2DM patients treated with metformin, there was a significant reduction in levels of RAGE isoforms, β-amyloid, and inflammatory cytokines, indicating a decrease in oxidative stress and tissue inflammation [75].

Anti-diabetic treatments have also been shown to affect antioxidant function and levels. For example, paraoxonase 1 (PON 1) is an enzyme that primarily functions in the liver as an antioxidant against lipid oxidation [76]. In a study conducted using a rat model, metformin was associated with increased PON 1 activity in diabetic rats, increasing its protection against ROS [77]. However, previous studies regarding metformin and antioxidant capacity were unable to quite confirm its direct effects. While the previous study showed an increase in antioxidant activity, metformin has been shown to instead prevent antioxidant levels from increasing. The study theorized that this could be due to metformin limiting the production of ROS in the first place, eliminating the need for increased antioxidant activity [78].

It is evident that metformin plays a protective role in protecting diabetic patients against oxidative stress by decreasing the production of harmful species and/or increasing antioxidant activity within cells. Nevertheless, further studies involving the direct effects of metformin as well as its specific mechanism of action are warranted.

## 7. Potential Therapeutic Strategies

The current antibiotics-based regimen for TBM treatment is the same used to treat pulmonary TB. This standard TB therapy regimen consists of a combination of isoniazid (INH), rifampin (RIF), pyrazinamide (PZA), and ethambutol (ETH) administered for 2 months, followed by INH plus RIF for another 4–10 months [37,79]. For these antibiotics to be effective against TBM, they must cross the BBB and reach the therapeutic concentration at the site of infection in the brain [10,38]. It was reported that RIF and ETH poorly cross the BBB, and limited studies are available on the anti-TB drug distribution in various compartments of the brain [10,37,38,80]. Additionally, the pharmacokinetic characterization of the anti-TBM drug to understand the efficient therapeutic concentration in the brain is yet to be determined [81]. Anti-inflammatory agents, such as corticosteroids, are often used in conjunction with anti-TB drugs to improve TBM treatment. However, the immunosuppressive nature of corticosteroids poses the risk of acquiring new infections and reactivation of latent TB. Therefore, new and improved therapeutic strategies are urgently needed for the efficient management of TBM [82,83].

Metformin is associated with reduced intracellular growth of Mtb in infected mice and enhanced efficacy of anti-TB drugs [84]. In Mtb-infected mice treated with both metformin and traditional anti-TB medications, including isoniazid (INH) or ethionamide (ETH), mice treated with metformin and INH were reported to have a lower bacterial load in their lungs compared to mice treated with INH alone [84]. Based on the discussion above, metformin used in combination with additional therapies to combat oxidative stress may have additional therapeutic value in the prevention and treatment of diabetic patients with TBM. Supplementing metformin with oral vitamin D, a nonenzymatic antioxidant, has been shown to further reduce advanced oxidation protein products [85]. However, it is important to note that metformin has been associated with respiratory side effects such as hypoxia secondary to lactic acidosis caused by metformin [86]. In T2DM patients with pulmonary conditions, such as chronic obstructive pulmonary disease (COPD), metformin was associated with increased bacterial pneumonia infections and usage of ventilators [87]. Therefore, when treating TBM, especially when symptoms of respiratory insufficiency are present, the risks should be carefully evaluated before treating with metformin.

Though metformin is the first-line treatment for T2DM, other medications, such as sulfonylureas, a-glucosidase inhibitors, and meglitinides, are also approved to act as glucose-lowering agents [88]. Pharmaceutical therapies that are specifically associated with decreased biomarkers of oxidative stress in animal models include, but are not limited to, thiazolidinediones, pioglitazone, and statins [89]. Statins are commonly prescribed to T2DM patients as preventative care since diabetes is a risk factor for the development of cardiovascular pathologies. In addition to managing lipid levels within the body, statins also possess anti-inflammatory properties. They have been reported to decrease inflammation via reduced C-reactive proteins [90]. In human endothelial cells exposed to hyperglycemic environments, atorvastatin and simvastatin have been shown to decrease production of ROS through the inhibition of HMG-CoA reductase in cholesterol synthesis [91]. Sodium-glucose cotransporter-2 (SGLT2) inhibitors can also be used to lower blood glucose levels by inhibiting glucose reabsorption within the kidneys. Similar to statins, they interrupt oxidative stress production pathways, leading to lowered ROS generation as well as reduced inflammatory biomarkers, such as AGEs [92]. Lower blood glucose and reduced oxidative stress markers have also been reported after vitamin C supplementation [93]. Overall, these pharmaceutical therapies have pleiotropic benefits that extend beyond their intended use. Statins and SGLT2 inhibitors were intended for lipid and glucose control and meant to reduce T2DM complications. However, patients can also benefit from their anti-inflammatory effects, leading to improved overall clinical outcomes.

High levels of H_2_O_2_ may stimulate further production of ROS; hence, there is a need to limit the levels of H_2_O_2_. GSH, in the presence of GPx, may be used to manage mitochondrial ROS production by reducing H_2_O_2_ to water [94]. Oral administration of the GSH precursors cysteine and glycine resulted in individuals with T2DM having restored rates of GSH synthesis and, in turn, reduced oxidative stress and oxidant damage plasma markers [58,95]. Considering that individuals with T2DM exhibit an associated reduction in levels of GSH, GSH supplementation may be an option to help combat oxidative stress and prevent TBM in these patients (Figure 4). 

Moreover, supplementation of metformin with oral vitamin D, a non-enzymatic antioxidant, has been shown to further reduce advanced oxidation protein products [85]. A spectrum of pharmacologic therapies has been associated with decreased biomarkers of oxidative stress in animal models, including thiazolidinediones, pioglitazone, and statins [89]. A reduction in blood glucose levels and oxidative stress biomarkers has also been reported following vitamin C supplementation [93]. Though fruits can increase glucose levels within the body, various studies have shown that sweet limes, oranges, apples, cranberries, grapes, pomegranate, and blueberries have been identified as harboring antioxidant properties and, thus, reduce oxidative stress levels among diabetic patients [93]. Furthermore, regular exercise can both reduce ROS production and enhance antioxidant capabilities, while also increasing insulin sensitivity, resulting in better glucose uptake into cells and lowering blood glucose levels [93]. Such exercises that have been shown to counteract the effects of oxidative stress are Tai chi, Pilates, and moderate-intensity aerobic exercises.

Hence, since oxidative stress is highly implicated in the pathogenesis of TBM, we propose that the aforementioned therapeutic strategies may offer protective benefits beyond Mtb lung infections and can prevent TBM, specifically in diabetic patients.

## 8. Discussion

We have discussed the association of TBM with DM in terms of oxidative stress and susceptibility to infection. Individuals with diabetes, particularly T2D, exhibit heightened levels of oxidative stress, leading to cellular damage and inflammation. This condition may compromise the ability of the immune system to fight Mtb infection, particularly in the CNS.

Whereas the host immune system generates ROS in the defense against exposure to Mtb, the overproduction of ROS in diabetics leads to tissue damage, including brain tissue, contributing to the development of TBM. ROS may also impact the integrity of the BBB—the major defense mechanism of the CNS. Damage to the endothelial cells lining the BBB increases permeability to pathogens and contributes to the development of TBM. Oxidative stress may also play a role in the accumulation and aggregation of Aβ peptides in the brain, an identified biomarker of TBM. Diabetes leads to an overproduction of ROS in addition to impaired antioxidant defense mechanisms, further contributing to a state of oxidative stress. Several factors, such as glucose auto-oxidation, glucose metabolism, and AGEs, contribute to the increased generation of ROS in individuals with diabetes. Moreover, the impairment of antioxidants, such as GSH, and antioxidant enzymes, such as GR, CAT, and SOD, worsen the effects of ROS and oxidative stress. The combination of increased ROS production and reduced antioxidant defense mechanisms creates a state of oxidative stress, leading to cellular damage and inflammation.

Considering the role of oxidative stress in the pathogenesis of TBM and the increased levels of oxidative stress in diabetes, we propose the use of antioxidants and other therapies targeting oxidative stress in the treatment of TBM. This article also highlights the role of metformin, a common diabetes medication, in reducing oxidative stress in diabetic patients. Thus, combining metformin with additional therapies, such as vitamin D or GSH supplementation, is a novel therapeutic strategy that may have significant value in combatting oxidative stress and improving treatment outcomes in diabetic patients and preventing the development of TBM. Overall, this review sheds light on the complex relationship between TBM and diabetes co-infection, with emphasis on the role of oxidative stress, suggesting that targeting oxidative stress through various therapeutic approaches could have beneficial effects in preventing TBM in diabetics.

Nevertheless, the limitations inherent in this review involve the fundamental nature of literature reviews, which inherently rely upon previously published studies and their accessibility. Specifically, we encountered a lack of comprehensive data pertaining to the direct relationship between TBM and DM. One anticipated challenge within the potential therapies proposed in this review is the establishment of a non-human model for the purpose of investigating the coexistence of DM and TBM to study the effect of various therapies. Moreover, the prospective therapeutic interventions suggested for mitigating oxidative stress, notably the employment of metformin, predominantly draw upon evidence derived from animal models, and a paucity of human-related data persists. These gaps in the current knowledge, pervasive within the available literature, inevitably constrain the capacity to offer efficacious and pragmatic potential therapeutic strategies. Thus, there exists a need for additional research in this field to build upon the existing findings and contribute to the expanding body of evidence that supports the connection between oxidative stress and the co-occurrence of TBM in individuals with diabetes.

## Figures and Tables

**Figure 1 biomedicines-11-02568-f001:**
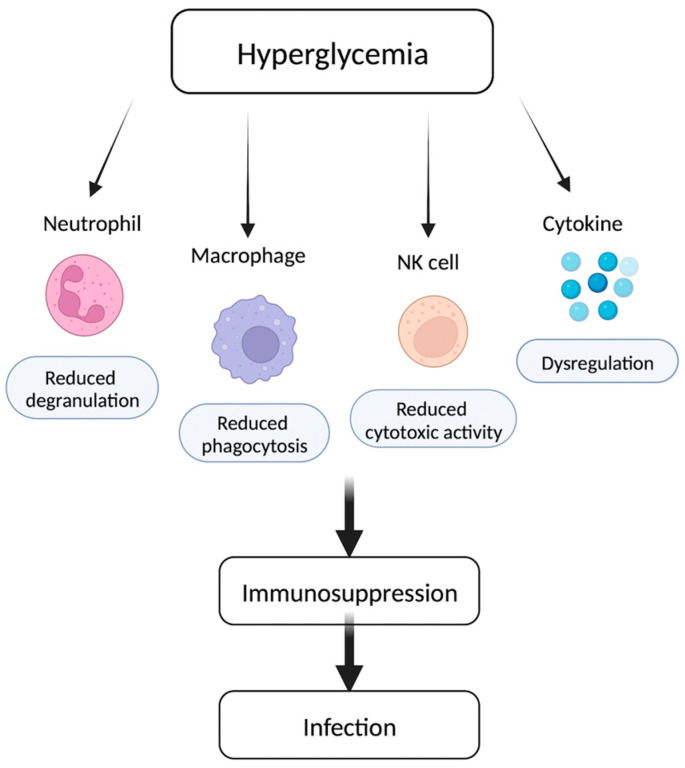
As blood sugar levels rise, the immune system operates with less efficiency, leading to increased vulnerability to infections, such as *Mycobacterium tuberculosis* (Mtb). Neutrophils and macrophages, crucial for combating infections, exhibit restricted degranulation and phagocytic activity in individuals with diabetes. Hyperglycemia contributes to a decline in natural killer (NK) cell functionality, resulting in diminished cytotoxic activity. Additionally, diabetic individuals experience cytokine dysregulation, which gives rise to a state of chronic inflammation.

**Figure 2 biomedicines-11-02568-f002:**
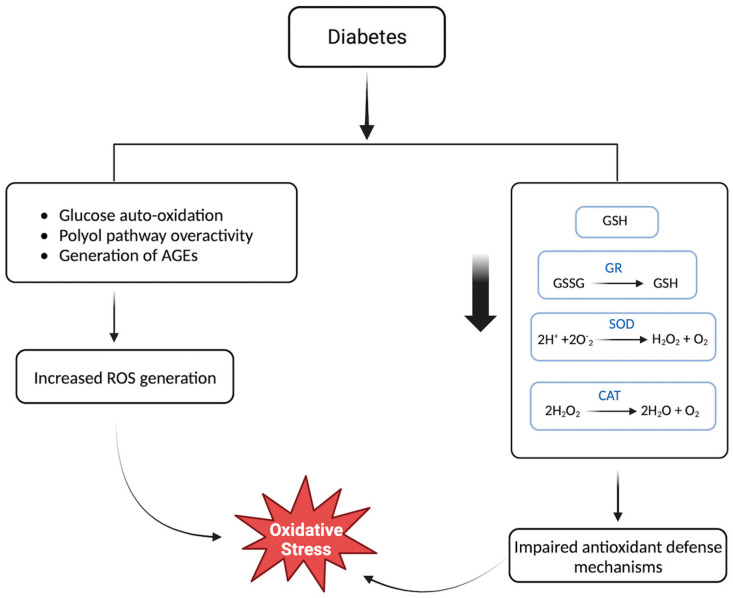
Hyperglycemia in individuals with diabetes leads to an elevation in glucose auto-oxidation, the activation of the polyol pathway, and the formation of advanced glycation end products (AGEs). These processes collectively contribute to an increased generation of reactive oxygen species (ROS), which subsequently results in increased oxidative stress. Furthermore, diabetes is associated with reduced levels of antioxidants such as glutathione, as well as diminished activity of antioxidant enzymes such as glutathione reductase (GR), superoxide dismutase (SOD), and catalase (CAT). The compromised functioning of the antioxidant defense system in diabetic individuals results in the accumulation of ROS, ultimately leading to increased oxidative stress and subsequent damage.

**Figure 3 biomedicines-11-02568-f003:**
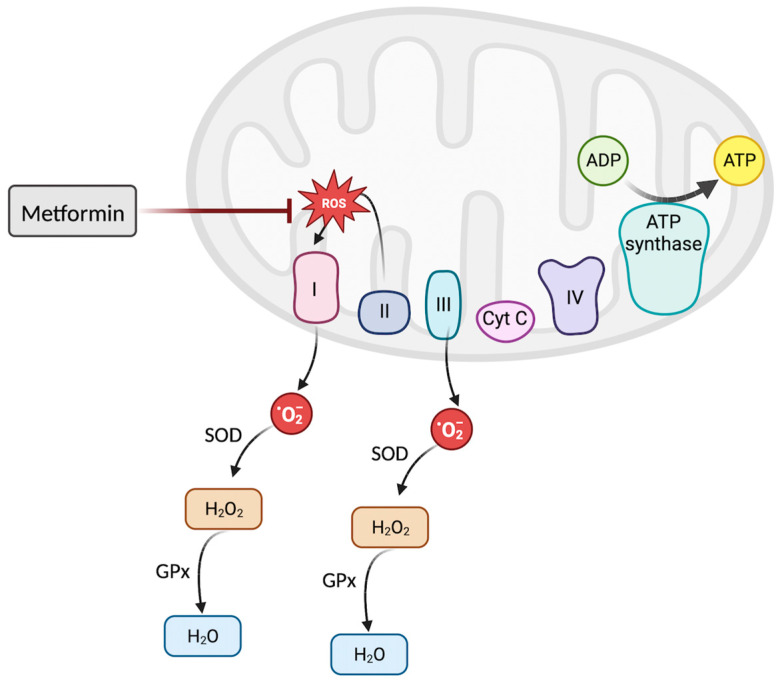
The electron transport chain (ETC) is composed of four complexes that collaborate to generate energy within the mitochondria. Typically, mitochondria produce reactive oxygen species (ROS) through complexes I and III during respiration. Nonetheless, the ETC may also function in reverse, allowing electrons that enter complex II to circulate back to complex I. This reversal leads to the generation of ROS. Metformin has been suggested as a potential inhibitor of this phenomenon, thereby potentially decreasing ROS production and aiding in the regulation of oxidative stress.

**Figure 4 biomedicines-11-02568-f004:**
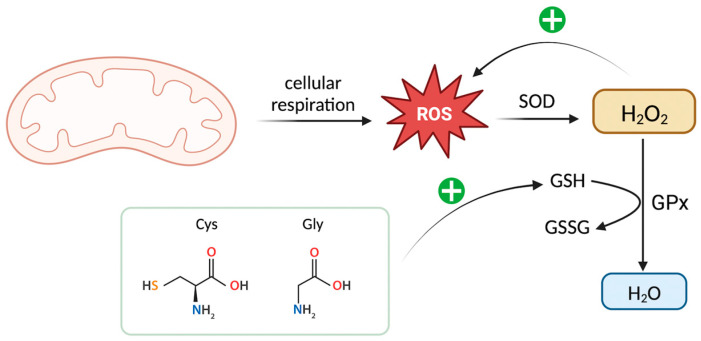
Reactive oxygen species (ROS) are generated by the mitochondria as a consequence of cellular respiration. In the presence of superoxide dismutase (SOD), ROS undergoes catalysis to form hydrogen peroxide (H_2_O_2_). Glutathione (GSH), aided by the presence of glutathione peroxidase (GPx), functions to counteract H_2_O_2_ by converting it into water, thereby assisting in the regulation of oxidative stress. Nevertheless, when the levels of H_2_O_2_ become elevated, they can trigger the production of additional ROS, thereby intensifying oxidative stress within the cellular environment. Administering precursors of GSH, such as cysteine and glycine, aids in the restoration of GSH levels. As a result, this helps to manage H_2_O_2_ levels and subsequently control oxidative stress.

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
