# Peer review of "Role of Oxidative Stress in Tuberculosis Meningitis Infection in Diabetics"

_biomedicines, 2023, doi:10.3390/biomedicines11092568_

Round 1

Reviewer 1 Report

This is a very nice review summarizing the effects of oxidative stress in diabetes and how it predisposes diabetes patients to TB infection. I suggest the authors add a section on other anti-diabetes drugs (other than Metformin). Also, please comment on the role of lifestyle changes (diet and physical activity) in mitigating oxidative stress. 

Author Response

We appreciate your feedback on the manuscript and believe the suggestions made have significantly enhanced the content of the manuscript.
Specifically, the inclusion of a section discussing anti-diabetes therapies beyond Metformin, such as sulfonylureas, alpha-glucosidase inhibitors, and meglitinides, adds valuable depth to the discussion.
Additionally, the expanded exploration of the role of lifestyle modifications in mitigating oxidative stress in diabetic individuals contributes further to the comprehensiveness of the manuscript. Thank you for your diligent work in improving the document.

Reviewer 2 Report

I read with great interest the article titled " Role of Oxidative Stress in Tuberculosis Meningitis Infection in Diabetics" by Navasardyan et al.

The paper's design is sound, and the article is logically organized into appropriate sections and subsections. English is generally fine, only minor spell check needed.

Here are the comments and suggested revisions:

1.      Oxidative stress and diabetes section: This should be expanded more and should be also underlined the role of inflammation, as it Is a key point for both diseases. You can use this recently published paper (doi: 10.3390/cimb45080420). In addition, by modifying so would further enhance the metformin anti-inflammatory effect which is reported after.

2.      The only problem of metformin use is associated with acute respiratory insufficiency, which could in same cases be a clinical manifestation of TBM. The authors should point it out in the potential therapies.

3.      Moreover, when we are dealing with type 2 diabetes, it is always better to have a multifactorial approach, as it not only ameliorates the oxidative stress, but improve patients’ outcomes (doi: 10.1186/s12933-022-01674-7). In facts, also other medications used in type 2 diabetes treatment, such as statins, also present anti-inflammatory effects which can in turn further enhance the amelioration of the clinical outcome and of the oxidative stress production (doi: 10.1097/FJC.0000000000001041).

4.      What are the limits to what reported and especially of the potential therapies proposed?

Author Response

Reviewer 2 provided wonderful feedback and suggestions regarding the gaps in the manuscript. We have revised the manuscript with these suggestions and believe it has thoroughly improved the manuscript.
1. In lines 299-317 we have added an additional section on the role of inflammation in oxidative stress and diabetes per the reviewer’s recommendation. This section discusses the role of sustained ROS production in diabetes due to increased inflammation, leading to cell damage and reduced ability of the antioxidant defense system in mitigating damage.
2. In lines 391-398 we have added discussion on the role of metformin in respiratory insufficiency in some cases, highlighting the need for caution when treating individuals with respiratory disease.
3. In lines 404-420, we have substantially revised the manuscript to discuss a multifactorial approach and have added a discussion regarding the use of statins.
4. In lines 493-506, we have revised the manuscript to include the limitations of the current review, including the limitations of the proposed therapies. We agree that this was a necessary addition to the paper given that there exists certain gaps in the literature.

Round 2

Reviewer 2 Report

The paper has much improved and can be further process for publication